# Identifying Factors to Facilitate the Implementation of Decision-Making Tools to Promote Self-Management of Chronic Diseases into Routine Healthcare Practice: A Qualitative Study

**DOI:** 10.3390/healthcare11172397

**Published:** 2023-08-25

**Authors:** Nina Sofie Krah, Paula Zietzsch, Cristina Salrach, Cecilia Alvarez Toro, Marta Ballester, Carola Orrego, Oliver Groene

**Affiliations:** 1OptiMedis, Research and Innovation, 20095 Hamburg, Germany; 2Avedis Donabedian Research Institute, 08037 Barcelona, Spain; 3Faculty of Management, Economics and Society, University of Witten/Herdecke, 58455 Witten, Germany

**Keywords:** self-management interventions, decision aids, implementation research

## Abstract

This study, as part of the COMPAR-EU project, utilized a mixed-methods approach involving 37 individual, semi-structured interviews and one focus group with 7 participants to investigate the factors influencing the implementation and use of self-management interventions (SMIs) decision tools in clinical practice. The interviews and focus group discussions were guided by a tailored interview and focus group guideline developed based on the Tailored Implementation for Chronic Diseases (TICD) framework. The data were analyzed using a directed qualitative content analysis, with a deductive coding system based on the TICD framework and an inductive coding process. A rapid analysis technique was employed to summarize and synthesize the findings. The study identified five main dimensions and facilitators for implementation: decision tool factors, individual health professional factors, interaction factors, organizational factors, and social, political, and legal factors. The findings highlight the importance of structured implementation through SMI decision support tools, emphasizing the need to understand their benefits, secure organizational resources, and gain political support for sustainable implementation. Overall, this study employed a systematic approach, combining qualitative methods and comprehensive analysis, to gain insights into the factors influencing the implementation of SMIs’ decision-support tools in clinical practice.

## 1. Introduction

It is well-documented that the healthcare sector has a poor record for the adoption of innovations [1,2]. The healthcare sector is particularly slow in adopting Information and Communication Technologies, a feature typically ascribed to human and organizational factors [3]. Scholars of diffusion of innovation in healthcare have documented the inherent complexities in the spread and adoption of innovations, detailing both push and pull factors and exploring the role of evidence in driving professionals’ adoption of innovation. The summary of this is clear: “scientific evidence is important but is not sufficient in itself to ensure that an innovation diffuses into practice” [4].

These consistent findings underline the fact that developing and sharing evidence-based decision tools is not in itself sufficient to ensure their adoption. There is a clear need to explore how their implementation can be ensured in practice. Healthcare settings are prone to implementation challenges, given the autonomy of the medical profession and complex hierarchical structures. Middle management has an important role in the implementation of healthcare innovations, which in turn are influenced by top managers [5]. Both mid-level and top management levels interact with the self-governing parallel structures of medical work. In addition, decision-making tools aiming at improving the patient journey across the whole patient pathway require the collaboration of—or at least alignment of—organizational processes between organizations such as hospitals and primary care centers. Previous studies highlighted the particular roles and perceptions of doctors, nurses, and managers in making decisions to adopt healthcare decision aids [6,7]. Awareness of the evidence of the potential impact on patient care and efficiency, as well as opportunities for system integration, are factors frequently identified. In addition to expected barriers such as costs, learning curves, IT integration, usability, and literacy requirements, studies also indicate that implementation is possible and can add substantial value to both patient care and managerial efficiency [8,9].

This is particularly the case in the context of self-management interventions, which extend beyond the actions of individuals or single organizations. Self-management interventions (SMIs) are supportive interventions aimed at increasing patients’ skills and confidence in their ability to manage long-term conditions [10]. Self-management interventions can be characterized in relation to the intervention characteristics (e.g., support techniques, delivery methods, provider type, location, recipient), target population, expected self-management behaviors (e.g., lifestyle behavior, clinical management, psychological management, social management, working with health or social care providers) and in relation to outcomes of SMIs (including empowerment, adherence, clinical outcomes, quality of life, perceptions/experiences, health care utilization, or costs) [11]. In the COMPAR-EU Project we comprehensively assessed the evidence of self-management interventions and developed a series of decision-aids and implementation tools (Box 1).

Given the pressure on healthcare systems through the rise of chronic diseases, which require effective self-management, the implementation of self-management interventions across the full patient pathway is of paramount importance. For SMIs, an implementation model which is suitable for primary care may not be appropriate for hospital settings, and a model focused purely on HCPs or on managers is too limited. A mixed approach is required, involving different information gathered from different types of organizations. This includes hospitals and community-based providers who provide care and support to patients with relevant chronic conditions.

Box 1The COMPAR-EU Project [12]COMPAR-EU is a multimethod, interdisciplinary project that contributes to bridging the gap between current knowledge and
practice of self-management interventions (SMIs). COMPAR-EU aims to identify, compare, and rank the most effective and cost-effective self-management interventions (SMIs) for adults in Europe living with one of the four high-priority chronic conditions: type 2 diabetes mellitus (T2DM), obesity, chronic obstructive pulmonary disease (COPD), and heart failure. The project provides support for policymakers, guideline developers, and professionals to make informed decisions on the adoption of the most suitable self-management interventions through an IT platform featuring decision-making tools adapted to the needs of a wide range of end users (including researchers, patients, and industry). COMPAR-EU launched in January 2018 and was completed in December 2022, contributing the following outputs: (i) an externally validated taxonomy composed of 132 components, classified in four domains (intervention characteristics, expected patient (or carer) self-management behaviors, type of outcomes and target population characteristics); (ii) Core Outcome Sets (COS) for each disease, including 16 outcomes for COPD, 16 for Heart Failure, 13 for T2DM and 15 for Obesity; (iii) extraction and descriptive results for each disease based on 698 studies for Diabetes, 252 studies for COPD, 288 studies for Heart Failure and 517 studies for Obesity; (iv) comparative effectiveness analysis based on a series of pairwise meta-analyses, network meta-analysis (NMAs) and component NMAs (CNMA) for all outcomes across all four diseases; (v) contextual analysis addressing information on equity, acceptability and feasibility; general information on contextual factors on the level of patients, professionals, their interaction and the health care organization for those interested in implementation; (vi) cost effectiveness conceptual models have been created for each chronic condition including risk factors or intermediate variables relevant for SMIs and final outcomes; (vii) business plans and a sustainability strategy was developed based on a multi-prong approach including qualitative interviews
with managers and clinicians, the focus group with clinical representatives from EU countries, workshops with industry representatives and a hackathon event.The majority of the COMPAR-EU end-products are available on the online COMPAR-EU platform: www.self-management.eu (accessed
on 29 June 2023).Watch the introductory video about the decision aids: https://youtu.be/_nqy6s79ZcY (accessed on 29 June 2023) 

With the aim of exploring how self-management-decision tools can be implemented into routine healthcare settings, ensuring effective use of evidence on SMIs, this study aimed to investigate the implementation factors for a specific suite of SMI decision aids from the perspective of healthcare decision-makers and professionals in hospital settings and primary care.

## 2. Materials and Methods

To make the evidence of SMIs available and understandable for different stakeholders (clinicians, policymakers and researchers, and patients), the COMPAR-EU project developed an interactive platform including three types of decision-making tools based on the GRADE approach (Grading of Recommendations Assessment, Development, and Evaluation), a method to assess the certainty in evidence and strength of recommendations:Interactive Summary of Findings tables (iSoF): these presentations will provide information in different formats about the quality of evidence and magnitude of relative and absolute effects for each of the core outcomes identified;Evidence to Decision frameworks (EtD): using semiautomatic templates, interactive EtD frameworks will be completed for a number of priority questions that will take into account the magnitude of desirable and undesirable effects, stakeholder views on the importance of different outcomes, information on resource use and cost-effectiveness, impact on equity, and other aspects like acceptability or feasibility of the interventions. The frameworks include draft recommendations that could be then applied or adapted to different settings;Patient Decision Aids (PtDA) were developed in plain language for all selected situations identified in the previous phases of the study. The aids were produced in six languages (English, French, German, Spanish, Dutch, and Greek) and included evidence to guide decision-making toward patient needs.

### 2.1. Study Design

This study has an explorative qualitative mixed-methods design using semi-structured interviews with decision makers (DMs) and health care professionals (HCPs) from Germany and Spain and conducting a focus group with DMs and HCPs from other COMPAR-EU countries. The design is based on a protocol developed for this specific design and published a priory in Open Science Framework (OSF) [13]. This publication includes further background information on the rationale, design choices, sampling, and analytical strategy.

### 2.2. Setting, Sample, and Recruitment Process

The study was carried out between March 2022 and October 2022. We sampled interviewees from Germany and Spain for maximal contextual variation (with different health system organizational and purchasing contexts for SMI implementation).

As background work, we conducted a review of governance and accountability systems to identify organizational enablers. In this process, we identified Germany and Spain amongst the countries participating in the project as those with rather distinct governance and accountability systems (for example, in terms of health system financing, provider organization, payment systems of doctors, and patient registration) that allow investigating maximum variation with regard to SMI implementation factors. As it was not feasible to conduct this large number of interviews in all countries participating in the COMPAR-EU project, Germany and Spain were therefore chosen as settings for this study. The background to this assessment and details of the sampling approach are described in more detail in the Open Science Framework protocol [13].

Interviewees were sampled with regard to country, institution (hospital vs. primary care), experience with chronic care management, position (decision-maker vs. professional), and age and gender. The focus group included seven participants from other European countries: Netherlands (n = 2), Greece (n = 2), Belgium (n = 1), Czech Republic (n = 1), and Portugal (n = 1). The focus group comprised the same professional groups of HCPs (n = 4) and DMs (n = 3) and settings as in the interviews. Focus group participants were recruited internationally from other COMPAR-EU countries, namely Greece, Belgium, and the Netherlands, were planned to include 5–7 people per group (with 10 invited per session with the assumption that not everyone will attend), consisting of representatives where age and gender should be a balanced mix and who possessed a good command of English. Interviewees were recruited through a specialized agency. Focus group participants were recruited by the COMPAR-EU project partners through local contacts and existing panels of respondents.

### 2.3. Data Collection and Data Management

We conducted interviews with German subjects in German, interviews with Spanish subjects in Spanish, and the multi-country focus group in English. Two researchers from the respective partners in Spain and Germany conducted the interviews. The focus group was conducted by a German partner. Both the interviews and the focus group were held online via Zoom. The participants received information about the project, a declaration of consent, and a 6-min video about the decision tools via email before the interview. In addition, all participants were shown a brief video about the three types of decision-making tools that were developed within the COMPAR-EU Project: Interactive Summary of Findings tables (iSoF), Evidence to Decision frameworks (EtD), Patient Decision Aids (PtDA), illustrating the use of these tools on the COMPAR-EU web platform.

An interview guide was developed, including open-ended questions (Appendix A). The content and structure were guided by the Tailored Implementation for Chronic Diseases (TICD) framework and a realist review [14]. The interview guide was divided into ten parts, which were framed with introductory and concluding questions. The guide concluded with questions about the most important implementation factors of decision tools as well as the need for their use in the future in the healthcare system. As translating the interview guide into different languages is considered a difficult task because interviewers from different countries may have different views and experiences [15], the international research team met several times to adapt and translate the interview guide to ensure the cultural relevance of the questions and common understanding between both teams. We report on our methodological approach according to the COREQ Checklist [16].

### 2.4. Qualitative Content Analysis

A qualitative-directed content analysis (QCA) based on the work of Hsieh and Shannon [17] and Gale et al. [18] was conducted. We deductively developed a coding system based on the TICD framework [14]. The coding system was inductively refined by including codes emerging from the interviews. Data analysis was conducted in the local language, and results were translated into English and reported back to both researcher teams. Each research team chose an appropriate analysis tool. The German team used MAXQDA2020 analysis software, while the Spanish team applied the NVivo 20 software (as licenses for these tools were available to the project partners). The results of the analysis were discussed in regular team meetings with anchor examples from both countries. Anchor examples from each country were translated into English and compiled in an Excel document. Each research team was responsible for the quality of the translation.

To achieve a structured approach for the analysis, the researchers followed a guideline of 16 steps for direct QCA developed by Assarroudi et al. [19]. The 16 steps are the synthesis of the suggested methods of Hsieh and Shannon [17], Elo and Kyngäs [20], Zhang and Wildemuth [21], and Mayring [22]. The steps were divided into three phases: 1. the preparation phase, 2. the organization phase, and 3. the reporting phase (Figure 1).

In the first phase, the interview guide was developed, and the interviews were conducted and transcribed by edited verbatim transcription, i.e., word-by-word transcription edited for readability and clarity.

In the second phase, we followed three coding cycles: First coding cycle: Both research teams pretested the deductive initial coding system by analyzing two interviews independently. Each team discussed the new inductive codes on their own and set some coding rules. After that, both teams from Spain and Germany discussed which inductive codes should be included and agreed on general coding rules for further analysis. 2nd coding cycle: Both research teams pretested the extended coding system by analyzing two more interviews independently, i.e., interviews other than those analyzed in the first cycle. Each team discussed the new inductive codes on their own and checked the intercoder reliability. After the discussion, researchers coded the remaining interviews and highlighted those quotes that did not match any code of the coding system. Again, the teams discussed new codes. At this stage, we specified anchor examples for each code. Third coding cycle: Each transcript was revisited for an iterative third cycle, and the new and existing codes were applied until no new themes or concepts emerged. At this stage, we checked the results for consistency.

In the third phase, we summarized the main message for each code based on all quotes assigned to it. For each code, we also extracted two representative quotes, i.e., one from DMs and one from HCPs. The selected quotes were translated into English.

The focus group was conducted to contextualize the results from the interviews in Germany and Spain with a broader panel, including participants from other countries. They were not analyzed at the same level of detail; rather, headline findings were summarized with a focus on whether divergent views emerged from the focus group compared to interviewee-reported findings.

## 3. Results

Our analysis included a total of 37 interviews. A total of 20 were held in Germany and 17 in Spain. The interviewees were divided into two groups: HCPs and DMs. A broad spectrum of different healthcare organizations was represented in the sample, with the majority being hospitals (n = 19, 51%). Male (n = 19, 51%) and female (n = 18, 49%) participants were evenly matched. Their age ranged from 32 to 65 years, and their healthcare work experience from 5 years to more than 30 years. The average duration of the interviews was 51 min, with a range of 36–65 min (see Table 1).

We developed a coding system with five dimensions, 17 subdimensions, 50 codes, and 21 subcodes from this analysis. In total, 1591 text segments were assigned to the coding system. The key findings were structured in the five main dimensions of our coding system: 1. factors of decision tools; 2. individual healthcare professional factors; 3. factors of interaction; 4. organizational factors; and 5. sociopolitical and legal factors.

### 3.1. Factors of Decision Tools

#### 3.1.1. Use of Evidence

The participants pointed out that reliance on evidence is inevitable in clinical practice. While most DMs admitted to searching for scientific evidence only on demand for certain patients, HCPs stated that it is imperative to stay up to date on evidence-based medicine before making therapy decisions. Therefore, they were advised to attend clinical sessions and revisions of scientific evidence using clinical guidelines and publications. Most of the participants described accessibility to scientific evidence as an ongoing and easy process:


*Well, for me, I think it’s easy, because of my clinical experience and years of work, you discard what you know does not have the strength of evidence and go to the consensus or recommendation system. […]. And well, I know the sources of evidence to use.*
(HCP 14; hospital; Spain; Row 202)

Interviewees used databases from scientific societies, high-impact journals, online libraries, clinical trials of the pharmaceutical industry, and training from the corporate website as a source. Some DMs stated that accessibility to suitable evidence can be challenging as it involves a lot of research and is very time-consuming. German HCPs pointed out that they use their own and their colleagues’ experiences from team discussions to stay informed about the new evidence.

#### 3.1.2. Existing Patients and Target Group of Patients

DMs reported that most patients in primary care suffer from chronic diseases, whereas in hospitals, only a quarter of patients are chronically ill. All interviewees pointed out that the use of SMI decision tools is especially suitable for younger patients rather than older ones because patients have a greater affinity for technology; they are more familiar with the internet and have better access to it. According to the interviewed DMs, decision tools are especially suitable for the following patient categories: chronically ill patients with low comorbidity, patients with only one main diagnosis, introverted patients, patients with language barriers, patients who are between 30–50 years old, patients with a higher educational level and those who want to take action and improve their own well-being. In contrast, the use of decision tools would be less suitable for patients with low income, lower education levels, less internet literacy and access, patients above 65, and insufficient health literacy.

#### 3.1.3. Use of Decision Tools

Many interviewees were not familiar with SMI decision tools by the time of the interview:


*I don’t really know of any decision-making aids from my everyday life that would go in that direction.*
HCP 19; hospital; Germany; 7)

While the German participants mentioned that they distribute flyers with treatment and therapy options to their patients and forward them to self-help groups, the Spanish participants use peer groups, motivational interviewing, patient empowerment, or quality-of-life questionnaires to involve their patients in therapy decisions. One German DM from primary care referred to the decision aid Arriba, and another one noted that he uses TheraKey Diabetes from BERLIN-CHEMIE and a self-developed decision tool.

According to the participants, decision tools would have to show a clear improvement in patient care and in the achievement of goals for patients and clinic staff:


*Prove that ultimately significant improvement in patient care and improvement in goal achievement, that’s point one for me, for ultimately putting that in.*
(DM 15; primary care; Germany; 99)

Interviewees believed that the tools were suitable for primary care in chronic diseases, patient empowerment, patient well-being, improving patient health, discovering the best interventions, and sharing these interventions with patients. In addition, HCPs confirmed that decision tools help empower patients and clinicians to improve follow-ups with their patients.

Regarding the technical usability of the COMPAR-EU tools, some participants mentioned that the design needs to be friendly, intuitive (i.e., easy to use), and time efficient. They demanded that the tools need to be accessible for patients who are not digitally affine.

Most of the participants stated that the SMI decision tools were appropriate for use in primary care. It was also mentioned that university outpatient clinics could implement these decision tools. Few participants said that decision tools could also be applied in hospitals and support physicians to look at the evidence of self-management tools in a structured way to support patients at discharge, offering an opportunity for integrated care:


*“I think leadership must be shared in this moment. I mean, in the hospital you have the head of a service, or the one who knows the most about that disease, which are units, but the patient comes from primary care, and that is, it’s been my mantra for many years. We are here to help primary care and collaborate with them because they are the ones responsible for the patients.”*
(DM 11; hospital; Spain; 247)

### 3.2. Individual Healthcare Professional Factors

#### 3.2.1. Knowledge and Skills

Participants’ perceptions of their own knowledge regarding decision tools correlated closely with varying experiences in their work field, their engagement in their professional bodies, and their leadership responsibility within their healthcare institution. Participants who described themselves as highly engaged reported having experience with self-management programs or decision tools:


*“Oh, you know, I’m a chamber chairman in the district and my hobby is continuing education, continuing education of my colleagues […]. So I’m relatively fit, I get a lot of input.”*
(DM 12; primary care; Germany; 7)

DM and HCPs were reported to be aware of self-management measures for chronic conditions. While German DMs stated that they conduct shared decision-making (SDM) based on their experience and medical guidelines, German HCPs stated that they use similar tools in so-called “Disease Management Programmes”. Spanish HCPs pointed out that they are aware of existing decision tools but do not use them in their daily practice.

Participants‘ perceptions of their own practice represented their opinion that chronic conditions are best-managed long-term through interventions that involve patients themselves. Explaining all treatment steps and providing patients continuously with scientifically proven information were perceived as essential. However, HCPs believe that patients often simply accept what the doctor suggests. They showed doubts that patients could use decision aids properly.

In order to use decision tools efficiently, both the majority of DMs and HCPs claimed that it is important for clinical staff to understand the principle of decision tools in detail and the meaning of self-management measures, to know the needs of the patients and to be able to explain them convincingly:


*“So first of all, they have to be so confident that they know these decision-making tools and how to use them, whatever. That they can communicate that.”*
(HCP 14; primary care; Germany; 87)

In addition, it was mentioned that motivation, personal responsibility, sensibility towards the patients, and affinity for technology are essential skills for the use of the tools.

#### 3.2.2. Cognitions and Attitudes

DMs and HCPs saw decision tools for SMIs as an innovation. They stated that innovation means changes, new working processes, and often resistance from HCPs:


*“I think it would be an innovation. So, it’s nothing that you can’t imagine as a doctor or as a patient. Such a tool is actually obvious, but although it is an obvious measure, I don’t know of any directly comparable one that is in daily use. And in this respect it is something new.”*
(DM 17; hospital; Germany; 43)

The tools should, therefore, convince the users, bring real evidence-based value, and should significantly help in a therapy decision and save time in the process.

Participants considered effectiveness and perceived benefit in the workflow of decision tools to be success factors for implementation. While some participants did not see the added value of decision aids at the time of the interview, both German and Spanish interviewees expected decision tools to become more important in the future.

German and Spanish DMs emphasized their intention and motivation for the use of decision tools based on the effect on patient outcomes. Treatment successes and recognizable progress would increase the motivation to continue:


*“In other words, feeling supported and having a script for how to do things helps. Because, at the same time, it structures the intervention. And in that way it can serve to evaluate you, to evaluate how things work. I think it’s interesting and well, come on, it’s something I always believe in. It is a methodology in which I like to work like this. Have, well, a process and see how, next step, next step, evaluation and see how it works.”*
(HCP 4; hospital; Spain; 296)

Some participants experienced uncertainty about the efficacy in clinical practice because they assumed patients might not follow their advice. Some interviewees already used decision aids, but they failed or did not lead to success.

#### 3.2.3. Professional Behavior

According to most DMs, structured preparation of a doctor-patient discussion was perceived as essential for success. They believed that professional behavior consists of explaining several available options to the patient without overwhelming him. HCPs would assume a consultative role in which they would decide together with patients which treatment steps to follow next. They confirmed that the patient’s condition determines what they can discuss with the patient and entrust them to do. Some of the HCPs would also involve family members or caregivers.

Regarding their capacity to plan change by using decision tools, some respondents indicated that they have the time to use decision aids for patients because the responsibility lies with the patient, and the clinicians should only be companions on the patient’s healing journey. Others, however, were very critical of the capacities for decision support in healthcare institutions such as hospitals and primary care centers, arguing that there is no time and no financial incentive for it.

Some interviewed DMs were also self-critical. They mentioned that sometimes they had no time to assess the current evidence, and decision-aid implementation projects had failed (DM 12; primary care; Germany; 9). Additionally, some were critical for not having tackled the aim of decision tools thoroughly enough. Some of the interviewed HCPs perceived the need to communicate with many different teams and characters in healthcare facilities as a challenge.

### 3.3. Factors of Interaction

#### 3.3.1. Interaction with Patients

Most interviewees stated that it is important to consider patient needs and characteristics (such as socioeconomic status, level of education, language skills, and access to digital devices) when implementing SMI decision tools.

Patients’ beliefs and knowledge can influence the way these tools are used. They need to be motivated to set achievable goals and by sharing positive experiences in group patient training:


*“[…] of course, people have to be a little bit interested in their own health. And be willing to change something. Because that also means a bit of work for them to register and take care of it. And yes, if they are not motivated, then it will be difficult. But I think that if they realize that they can change something and have a positive influence on the disease, then that is of course motivation enough.”*
(HCP 20; hospital; Germany; 31)

One of the crucial factors in the successful implementation of decision tools is patient preparation. Patients can be prepared before a consultation by providing an email link to the tools or after consultation by having a nurse or medical assistant explain the tools:


*“[…] it would be great if you could invite them [patients] directly to a small training session, for example. Or simply distribute flyers where videos explaining the procedure can be found.”*
(HCP 20; hospital; Germany; 66–69)

At the same time, there are roles and responsibilities carried out directly by the patients. Patients need to actively ask questions, take responsibility for self-management of their condition and show compliance with their treatment.

#### 3.3.2. Professional Interaction

Both DMs and HCPs stated that when it comes to professional interactions within the medical team, communication, information sharing, and experience need to be maintained by regular team meetings. One participant also mentioned that the HCPs implementing decision tools need to agree on fixed rules that they want to follow.

Within the team, team members need to develop a shared understanding of the benefits of the decision tools. All participants agreed that this could be achieved by showing the added value of the tools, emphasizing the evidence-based aspects of the tools, demonstrating positive experiences of other organizations, and highlighting patient benefits:


*“Our own colleagues from other hospitals, or another region, should explain to us the benefits that the tool brings. I think that that is the strategy we should follow. First, explain the purpose of the tool, then, have the experience of another place where we can see the health results that have been achieved with help of these tools. Show us the experience of patients that are using the tools […].”*
(DM 8; hospital and primary care, Spain; 270)

Additionally, several DMs from Spain suggested that healthcare organizations can establish an interdisciplinary group comprising administrative staff, social workers, physicians, and nurses that could oversee implementation and provide relevant support in implementing the tools.

Another important factor reported by the participants is building enthusiasm and support among the team members. Team members need to be involved in decision-making surrounding what kind of decision tools will be implemented in their organizations as well as in the processes of implementation of the decision tools. Further, enthusiasm can be increased by providing training and emphasizing that decision tools might reduce the team members’ workload.

The referral processes need to be maintained between HCPs and other team members, i.e., communication and coordination between different professionals (physicians, nurses, nutritionists, and psychologists) within the same organization as well as between different care levels (primary and secondary care). The referral processes between patients and HCPs can be maintained if patients are informed and guided by HCPs throughout the whole treatment. HCPs should monitor how patients are feeling about self-management and make sure they are still happy with the intervention.

Both groups agreed that physicians and nurses should be informed about and engaged in the development of decision tools when implementing them into clinical workflow. Most HCPs stated that information and engagement about decision tools need to be maintained right from the beginning of the treatment. However, some Spanish HCPs also emphasized that they would prefer to be informed first when methods of distributing and using the decision tools are already tailored to clinical workflows. Here, the top management plays a key role:


*“I think medical directors are those who need to know the most. For his medical background, they are in contact with the heads of service, they know all the scientific commissions that depend on the medical direction. […] they can explain to us what they want to do, why, what situation we are in and what we hope to achieve with it.”*
(DM 11; hospital; Spain; 312)

Participants pointed out that decision tools need to be presented to patients as early as possible during the diagnosis. In Germany, a portion of DMs mentioned that it is mandatory for hospitals to inform about self-management when patients are sent home and treated as outpatients. Here, decision tools could provide support.

More generally, participants stated that the processes of using decision tools need to be centralized and bundled beyond organizations so that digital interoperability at intersections with other systems can be achieved. This helps avoid working with several different tools unnecessarily, and it could help streamline daily processes.

#### 3.3.3. Roles and Responsibilities

There are different roles and responsibilities that various team members assume when implementing decision tools (especially patient decision tools), e.g., decision-makers, physicians, nurses, and administrative staff. DMs have both the responsibility of presenting the tools to those who will implement them in their organization and a broader leadership role. Physicians were seen by participants as those who are responsible for identifying patients who can benefit from the use of decision tools, checking the evidence provided with the tools, and answering open questions arising when patients use the tools. They can coordinate tasks within the team. Some HCPs in Germany stated that in hospitals, ward physicians are more likely to support decision tools and that senior physicians or physicians at the middle management level are less interested in being involved in implementing innovations. In contrast, young physicians are more willing to implement the tools, as reported by Spanish HCPs. Nevertheless, both the Spanish and German participating groups agreed that the general practitioner plays a very important role as he or she often has a much closer relationship with patients than the other physicians and can monitor patients in everyday life.

The tool introduction can be delegated to nurses or administrative staff. Both groups can explain the tools and guide patients through them before or after the consultation, send them a link to the tools, and upload and update the results of decision tools in the patient information system. However, administrative staff and nurses must not give medical advice about self-management interventions to patients. Other team members to whom the task of tool introduction could be delegated are nutritionists, study nurses, data managers, social workers, psychologists, and cultural mediators.

All in all, both HCPs and DMs mentioned that the implementation process is a shared responsibility of the whole team, and they should agree together if and what tools will be considered.

### 3.4. Organizational Factors

#### 3.4.1. Incentives and Resources

For the successful implementation of SMI decision tools, three important types of resources were mentioned by the interviewees that were needed but not always available in reality: time, financial, and personal resources. Both DMs and HCPs pointed out that patients need training on decision tools, which might be very time-consuming. However, there is only a limited amount of time in clinical consultation:


*“Often it only takes place between door and door due to time constraints. But if we had a little more time in the clinic to really have another discharge discussion with the patient, so to speak. That would also be a good moment to refer to such a program.”*
(DM 17; hospital; Germany; 9)

Spanish DMs claimed that there is a need for organizational change to provide time resources for introducing decision tools to patients properly.

Referring to personal resources, some participants claimed that there is a lack of personnel even though the staff limit per patient was raised. Most participants confirmed the need for leadership for the implementation of decision tools.

German participants believed that the use of decision tools is highly dependent on financial resources. If sufficient financial resources were available, the use of decision-support tools can be supported. Some Spanish DMs complained that it is difficult to receive financial resources, and some Spanish HCPs did not see the possibility of implementing such tools in primary care at the current time because of financial difficulties in their institutions.


*“I’m going to be very sincere; I think we are in a critical moment in primary care in all of Spain. I mean, right now we are time wasting, we have very few tools and very little time to tend to patients. […] It doesn’t have to do, maybe, with what you are asking, but you want to evaluate a strategy, where probably the system starts to break in a few years and we will do what we did 40 years ago, which is visit the patients for a few minutes and not do any of the prevention and health promotion.”*
(HCP 10; primary care; Spain; 165)

Participants had different opinions regarding financial incentives and disincentives. Some DMs argued that financial incentives can have a positive and motivational effect on the successful implementation of decision tools, as organizations need to receive reimbursement for the time spent implementing decision tools. However, others believed that financial incentives could have less or no effect because clinicians should focus on patient health. In addition, if there were bonus payments, clinicians would have to pay higher taxes for that. Most of the interviewed HCPs were convinced that financial incentives, e.g., a bonus payment or voucher, could be offered to increase the use of decision tools for clinicians to feel fulfilled professionally. They expected cost savings when implementing decision tools into routine healthcare practice.

German DMs explained that, in the German healthcare system, the use of decision tools is not included in the primary care reimbursement plan or in hospitals. There is also no compensation for the prescription of SMIs for patients. Some suggested the use of decision tools in special units like diabetic clinics, where they could be a part of a complex treatment, and the reimbursement would be made through daily flat rates rather than diagnosis-related group payment rates (DRGs). Additionally, new centers could be established that focus on treatment and follow-up questions regarding the self-management of chronic patients—similar to telemedicine heart failure centers in Germany. Others argued that in the primary care reimbursement model (EBM-System), it is possible to include a new EBM Code (fixed flat rate) for consultation, including decision tools or bonus payments. The use of decision tools could be a part of “Disease Management Programmes” or other new treatment programs for self-management and prevention that could be established in cooperation with healthcare insurance companies. Regarding the hospital reimbursement model (DRG-System), interviewees suggested including the use of decision tools in the DRG rate by increasing the case mix or introducing a new operation and procedure code (OPS in Germany). They highlighted that the reimbursement model affects the likelihood of using decision tools because HCPs’ activities are determined based on payments and not on time spent on patients or how meaningful it is for patients.


*“And I say the other quite brutal key in medicine is a reimbursement, so whether that is paid in some form or whatever. Whether that’s somehow times-, whether that’s reimbursed in some form so to speak yes. That is certainly something that would always be a trigger or a driving effect.“*
(DM 18; primary care; Germany; 36)

Most of the participants perceived that non-financial incentives for the use of decision tools could be created if a major improvement in patient care could be achieved or if clinicians saved time in patient consultations using decision tools. Another non-financial incentive might be a certificate for using decision tools. DMs denoted that not only the empowerment of patients but also of clinicians is very important. HCPs evaluated the improved collaborations between clinicians, new tasks, and responsibilities as an incentive, so this could help develop their careers further.

Participants emphasized a need for interoperability with other systems or applications that measure health care outcomes (blood glucose, blood pressure, weight). Some argued that decision tools should be integrated into the information system of the provider and the system should have a uniform interface. Furthermore, some German DMs emphasized the importance of homogeneity of the tools. Instead of offering many different tools to providers, there should be one tool for all types of patients. DMs requested different technical requirements for the use of decision tools, such as accessibility by phone, integrated videos for SMIs, and simple interface and navigation. HCPs believed that online consultation and a hotline for technical questions should be offered. All participants pointed out that the implementation of decision tools should happen with the patient in mind, and decision tools should be without content-based gaps for patients.

The participants emphasized the importance of providing continuous training to all professionals involved in the use of tools. They claimed that a continuing education system is required for all team members involved, such as nurses, medical assistants, physicians, or data managers:


*“I would present and do it, for example, as part of cardiology or internal medicine training events, quality circles, local congresses. So that’s how medical innovations get into use.”*
(DM 18; primary care; Germany; 62)

DMs and HCPs presented different opinions about the impact of the use of decision tools on other healthcare institutions. DMs held the view that information could be noted in the doctor’s letter and thus give other healthcare facilities an idea of the SMI status of the patient. HCPs hypothesized that this might save time.

#### 3.4.2. Capacity of Organizational Change

DMs believe that the use of decision tools in everyday clinical practice is a question of authority and the associated power of persuasion. Further, they claimed that authority and persuasiveness would have an impact on the subsequent use of the tool by other team members. They emphasized that the use of new tools is more efficient if it is implemented and presented by opinion leaders. DMs stated that leaders need to believe in the project promote and incentivize adherence of decision tools by showing that the change is for the better.


*“The middle management, which we call supervisors, have to always be in the know of anything that is being implemented, which doesn’t mean that they are the ones who take leadership in these tools, because a head of service or a middle manager, do have a very wide vision, and they have a lot of knowledge in management and activities management, and numbers, but that doesn’t always go hand in hand with leadership regarding implementation of new things.”*
(DM 12; hospital; Spain; 313)

Several HCPs pointed out that new rules, regulations, and technical requirements would have to be created for the implementation of decision tools. DMs referred to the corona pandemic, where regulations, rules, and guidelines have multiplied. Some feared that decision tools would be another major bureaucratic hurdle. Furthermore, some Spanish DMs criticized that there are overregulated health services with many workers and very bureaucratic and rigid management and coordination systems that hinder the optimal execution of regulations, standards, and policies.

Many HCPs mentioned that decision tools have a high priority because they could reduce the great time pressure in the context of more efficient work. On the other hand, DMs pointed out that decision tools cannot adapt to the clinical workload, and the time pressure in hospitals and practices is so high that they do not fit the required new processes:


*“No, I don’t think it’s a high priority for now. Let’s just say that we have enough to deal with the normal challenges of everyday life. In this respect, it always has to be critically questioned.”*
(DM 6; primary care; Germany; 39)

According to the participants, monitoring and feedback play an important role in the successful implementation of decision tools. They highlighted the importance of the continuous short- and medium-term measurement of patient outcomes to prove that the tool is useful and worth to continue using it. In terms of assistance for organizational change, interviewees emphasized that external support is needed in addition to internal support from the users of decision tools. German participants suggested that external support can be offered by pharmaceutical, management, or insurance companies, whereas Spanish participants focused on national or autonomic healthcare systems and also perceived patient organizations and initiatives as important assistance to initiate organizational change for the use of decision tools.

### 3.5. Social, Political, and Legal Factors

#### 3.5.1. Economic Constraints on the Healthcare Budget

DMs in Germany emphasized that there is not any budget for the implementation of decision tools, but the economic pressure in the healthcare system may pressure budget allocation to self-management tools in the future:


*“If [implementation of decision tools] demands costs, then that’s ultimately the responsibility of the healthcare system to implement that. In my opinion, the problem is that the healthcare system requests a lot of actions, but it is not accordingly supported.”*
(DM 15; primary care, Germany, 67)

Similarly, DMs in Spain mentioned that there are a lot of activities that are required by the public healthcare system but cause economic pressures. This means that although organizations could save some money for the implementation of decision tools, there are a lot of competing activities to which they need to allocate the budget.

#### 3.5.2. Contracts

In Germany, a number of DMs argued that decision tools could be included in the contracts of so-called “Disease Management Programmes” accepted by the Ministry of Health in the whole of Germany. They could also be included in contracts between specific insurance companies and providers. Having several contracts with various insurance companies, however, may have a negative impact on successful implementation. While German participants saw problems with multiple contracts, in Spain, DMs worried about the open tendering process. If a provider requests a budget, e.g., for the implementation of decision support tools, public procurement law requires an open tender procedure, regardless of the budget amount. This is often a very administration-heavy and time-consuming process.

#### 3.5.3. Legislation, Legal Issues, and Data Protection Policy

Currently, the use of self-management decision tools is not included in the German Social Code (SGB V). In order to integrate decision tools into the healthcare setting, the state/Ministry of Health needs to present them as a prescribed overarching statutory concept and request that HCPs use these tools:


*“The system is learning; artificial intelligence will certainly lead to them getting smarter. The databases will become larger. When we finally have electronic patient records, that will certainly be supported institutionally, perhaps also in our country. I think that evidence is coming from this area.”*
(DM 9; primary care; Germany, 19)

While cooperation with commercial companies may bring about further legal issues that require clarifications, a general legally accepted procedure of the use of decision tools might be easy to implement and provide a degree of security. Both DMs and HCPs in Germany stated that data protection policy could lead to difficulties in implementation, as it entails much discussion unless specifically prescribed by a regulating authority. Spanish participants did not comment on legislation, legal issues, or data protection policy.

#### 3.5.4. Influential People and Organizations

DMs mentioned several influential organizations that should be involved when implementing decision tools: hospitals, larger medical clinics that have successfully implemented decision tools; societies and associations of experts; Ministry of Health; relevant health care authorities; Federal Institute for Drugs and Medical Devices; and insurance companies and patient organizations. One German DM stated, however, that the inclusion of pharmaceutical companies might complicate the implementation because they might have more interest in economic aspects.

#### 3.5.5. Healthcare System

In both Germany and Spain, DMs mentioned that the use of decision tools is not considered during the performance evaluation of healthcare systems. The German DMs think that the use of decision tools could be one of the factors used to measure performance within the healthcare system. The use of decision tools could be measured, for instance, through patient satisfaction surveys or simply by asking the patients if they were referred to decision tools in both hospitals and primary care practices.

In Germany, DMs argued that the use of decision tools is currently less consistent with the recommended ways of working in the healthcare system as digitalization is progressing slowly, and the current systems are not yet designed for patient interaction of that kind. Spanish DMs saw this differently. They noted that the approach to implementation of decision tools is aligned with the implementation strategies of other initiatives that are currently in place. Most of the participants reported that decision tools should be prioritized in the healthcare system because patients should take more responsibility for their own health and be more involved in SDM with their HCPs. Another incentive for digital innovation is the economic pressure created by unnecessary hospital admissions and consultations and the bottleneck among clinicians:


*“Based on the introduction of the DIGA [Digital healthcare applications], the interest for digital applications in healthcare will ultimately increase. And I think that in a few years, that’s going to be a help tool, especially for patients with increasing medical needs, and the shortages in medical care […]. The tool means for patients a kind of shared decision which helps them to get their treatment or achieve their goal.”*
(DM 15; primary care; Germany, 100–102)

#### 3.5.6. Social Changes and Paradigm

Digital tools supporting SDM between clinicians and patients align with future visions about the healthcare system, and as such, decision tools could lead to social change. They could empower patients in their own healthcare autonomy:


*“[…] there is still a paternalist attitude from health professionals towards patients. Patients follow physicians and nurses advises. I believe the step forward regarding patients’ participation must be undertaken.”*
(DM 8; hospital and primary care; Spain; 297)

In addition, better use of data and methods such as artificial intelligence will lead to making these tools more efficient and precise and save time and resources for both patients and doctors.

#### 3.5.7. Perspectives of Managers vs. Health Care Professionals

Overall, both DM and HCP addressed similar themes in relation to the five dimensions of the coding system and were broadly in agreement with the key barriers and facilitators for implementation, in particular in relation to awareness and training of professionals; decision aids as an innovation factor, the need of patient preparation, and allowing for sufficient time to address the output of the decision aid with the patient. Differences emerged on various points, such as the appraisal of the underlying scientific evidence of decision tools, where HCPs demonstrate a higher level of familiarity as compared with DM. In terms of the effects of using decision tools, HCP appeared to be more concerned with short-term efficacy, whereas DM demonstrated more interest in the longer-term outcomes and positive side effects on efficiencies. In this context, DM also reflected more often on the role of the organization (hospital vs. primary care) leading the implementation of the tool. Finally, different views were put forward regarding the use of financial incentives: whereas HCP provided a mixed assessment acknowledging both potential advantages and disadvantages, DM was overall positive about opportunities to link the adoption and implementation of decision tools to financial incentives.

#### 3.5.8. Contextualization of the Results by the Focus Group

The focus group participants emphasized that, as a first step in the successful implementation of decision tools, it is important to identify a group of patients for whom decision tools are most interesting based on their willingness to change something about their condition. In that way, it is maintained that decision tools are introduced first to the patient group with the greatest likelihood of using them. Only then should decision tools be introduced for all other patients. Further, participants reported that using financial incentives, including the use of decision tools in medical guidelines, and offering certification for those who successfully implement them in their settings can help increase the use of decision tools amongst the HCPs. The funders involved in the implementation process can differ within countries based on their healthcare system. While involving insurance companies from the beginning seems to be the most effective approach in insurance-based schemes, in public schemes, new tools are often piloted first, and organizations then apply for financial support. Another implementation factor was the early training of medical students. When medical students are trained to involve patients in SDM, such tools are more likely to be implemented (Table 2).

## 4. Discussion

This analysis shows that decision tools, such as the three developed tools as part of the COMPAR-EU project, can support the use of evidence about SMIs in healthcare practice. The implementation of self-management decision tools represents a digital innovation that stimulates and requires change and rethinking processes at different levels: individual, organizational, and system. The lack of use of SMIs and decision tools in practice is not only due to limited resources in different healthcare settings but also due to limited knowledge about the effectiveness of these interventions and tools. The use of new tools is more efficient when they are introduced and presented by opinion leaders. Digital innovations such as decision support require organizational resources such as time, personnel, and budget on the one hand and the right financial and non-financial incentives on the other. This depends not only on the resources and incentives provided internally but also on the support of stakeholders such as management or insurance companies and the healthcare system itself. Furthermore, the implementation of self-management decision tools can increase the autonomy of patients in therapy decisions and thus contribute to the socially and politically promoted paradigm shift in the doctor-patient relationship.

Based on our results, there are—in general—no major differences between implementation factors in hospitals or in primary and secondary care. Major deviations were also not identified when comparing interview and focus group results. These exploratory results provide a further understanding of the facilitators of the implementation of self-management decision tools into healthcare practice.

This study builds on and aligns with other bodies of work examining the implementation of decision tools such as patient decision aids or other evidence-based tools [23,24,25,26,27]. Implementation is unlikely to take place if HCPs are not aware of the use of decision tools. Training HCPs to deliver decision tools is essential. HCPs need to recognize the added value and proven effectiveness of such tools (improving patients’ quality of life and supporting decision-making) before using them in clinical practice [28]. Involving the whole team, including physicians, nurses, administrative staff, and middle and top management, in the implementation of decision tools and conducting regular meetings to exchange experiences is also often referred to in other studies [8]. Our study confirms this point and builds on this by showing that delegating a portion of the tasks to nurses and administrative staff (e.g., leading a group training for patients) can increase the responsibility and attractiveness of the profession.

Tol-Geerdink et al. [27] found out that almost all patients would accept if decision tools were introduced on the day of their diagnosis. This fits well with the results of our study to encourage the use of decision tools right from the beginning, either through online tools or involving groups, to reduce the time required by physicians to address questions [26]. Other studies illustrate that lack of time is one of the most frequently cited barriers when engaging SDM [23,26]. At the same time, it was also shown that use of decision tools can save time when HCPs hand out the tools to patients to use at home [26,27]. Less complex decision tools provided in different versions based on health literacy and knowledge of patients can be used [11,14,25].

Our study illustrates that financial incentives for organizations might help implement decision tools for self-management. However, the literature shows mixed results on whether financial incentives have an impact on the behavior change of HCPs in providing self-management [23]. In addition, financial incentives might only achieve a short-term change [26]. Nevertheless, financial incentives can be impactful if introduced in a size big enough and in the whole system simultaneously [24].

External factors like national guidelines or regulations can support the implementation of SMI decision tools [11]. The emergence of national governance and guidelines is already seen as an important driver elsewhere. For instance, there are several NICE guidelines in the United Kingdom that recommend SDM supported by decision aids [28,29] and guidelines urging the use of SDM for prostate cancer in the Netherlands [30]. These might provide support for the implementation of SMI decision tools in the future.

Our study emphasizes that individual HCPs need to be aware of the added value of decision tools for both patients and HCPs. HCPs further need to prepare patients for the use of decision tools and encourage them to share their preferences about SMIs in medical consultations. Healthcare organizations need to show that the use of decision tools is one of their priorities. They need to provide a structure with time, financial, and personal support for teams to implement decision tools, including avoiding competing activities to be done at the same time. They also need to train teams to give them confidence in using decision tools, let opinion leaders explain their added value to their teams, and motivate them about their use. Governments, payers, and policymakers play an important role in providing incentives (financial or non-financial) and incorporating the use of self-management decision tools and the SDM approach in national guidelines and in already existing structures or programs for chronic patients. Additionally, they need to integrate working with self-management decision tools in the performance measurement of healthcare systems. They should support further development of such tools by encouraging data collection and the use of artificial intelligence. In that way, these tools can become continuously more efficient and achieve time and cost savings. Decision tool developers need to ensure that tools are accessible to patients with low health literacy but also provide opportunities for patients who want to learn more about SMIs. They also need to consider different levels of digital affinity of various groups of users and verify the interoperability of decision tools with other systems.

The 37 interviews evaluated in this study provided a suitable amount and quality of data, as the interviewees addressed all interview questions in an open, detailed, and focused manner. Since the hospitals, secondary care, and primary care practices were of different sizes organizational structure, and came from two different European countries and cultures, the answers to some questions differed in terms of positive or negative perceptions. After coding eight interviews, there were no more adjustments to the category system. We reached saturation with the current sample; however, increasing the number of interviews further might have led to some additional data enriching the current findings. A weakness of the current study is that the seven participants of the focus group were recruited internationally, and the group discussions were held in English; hence, non-native participants might have had inhibitions to participate in the debate or weaknesses in expressing themselves. Furthermore, the focus group was not transcribed and analyzed at the same level of rigor as the interviews. However, its purpose was to contextualize the interview findings rather than to provide detailed accounts of focus group participants’ views on the subject matter. In that view, the focus group successfully validated the identified implementation factors from semi-structured interviews.

## 5. Conclusions

The aim of this study was to identify factors of successful implementation of self-management decision tools in routine healthcare settings. This study identified the main facilitators who can guide those who are willing to implement decision tools in their organizations. The results of this study can be used to develop business plans focusing on evidence-based decision tools, thus ensuring research exploitation. In the future, different versions of business plans need to be adapted if there will be differences applicable to different health systems and provider types. The results will not only contribute to the development of an implementation strategy for decision tools but also increase the empirical evidence about the use and transferability of innovations in health information systems.

## Figures and Tables

**Figure 1 healthcare-11-02397-f001:**
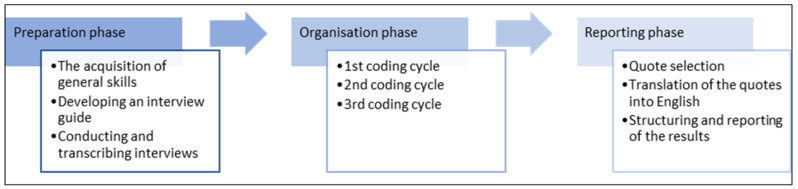
Flow chart of directed qualitative content analysis based on Hsieh and Shannon [17], Elo and Kyngäs [20], Zhang and Wildemuth [21], and Mayring [22].

**Table 1 healthcare-11-02397-t001:** Demographics of interviewees (n = 37).

Demographics	Categories	n (%)
Country	Germany Spain	20 (54) 17 (46)
Gender	female	18 (49)
	male	19 (51)
Age	32–45	11 (30)
	46–55	12 (32)
	56–65	14 (38)
Role	healthcare professional	17 (46)
	decision-maker	20 (54)
Institution	hospital	19 (51)
	special care	4 (11)
	primary care	14 (38)

**Table 2 healthcare-11-02397-t002:** Results of the focus group.

Factors	Key Topics from the Focus Group Discussion
Factors of decision tools	- Tools to be adapted to the different needs of patients as there are different levels of health literacy and digital literacy; - Patients to receive guidance to use tools and understand terms and/or navigation; - Accessibility via mobile app.
Factors of interaction	Role of clinical leaders - Health care professionals (HCPs) have one of the most important roles: they can promote and introduce patient decision tools to patients. Preparation by patients - Patients should be made aware of the decision tools in advance, before consultations—for instance, remotely (similar to Sweden/USA); - Patients need to be educated about the purpose, benefits, and use of decision tools; - Patients should be trained to ask questions regarding the summary from PtDA, as physicians might often forget about the Building enthusiasm and support - Consider the culture of the provider, find out if the organization is already considering shared decision-making, and use decision tools; - Start discussions about the importance of decision tools in practice within the team; - Build a working group that will be closely focused on the implementation of decision tools; - Show the evidence of the decision tools and how decision tools can be beneficial for both patients and clinicians.
Individual healthcare professional factors	- [were not addressed by focus group participants]
Organizational factors	Key: Alignment with organizational priorities - Belgium: it is not a priority of organizations; many do not know the term “shared decision making” or “patient decision aids”; - Greece: there are some first steps to implement self-management interventions (COPD: patients get spirometer for home); - Czech Republic: payers have an increased interest in evidence-based digital solutions and telemedicine. Incentives - Netherlands: no direct incentives to support SMI tools available; - Portugal: rewards for the family unit might be facilitated by SMI tool implementation; - Cave—financial incentives might have a misleading effect as HCP might take advantage of the system; - The use of decision tools could be implemented best in the clinical guidelines (also top-down approach) and/or in accreditation systems;
Social, political, and legal factors	- Highly relevant because chronic conditions cause an increased burden on the population, and patients need to be empowered to take an active role as they are the experts on their own health (clinicians are experts of medical support); - With better use of data and better methods (such as AI), decision tools will also improve in usability and precision.

## Data Availability

Data (transcripts) from the COMPAR-EU project have been deposited according to the data management plan at NIVEL. The qualitative datasets used and/or analyzed during the current study are available from the corresponding author upon reasonable request, provided they do not identify interviewees.

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
