# Peer review of "Identifying Factors to Facilitate the Implementation of Decision-Making Tools to Promote Self-Management of Chronic Diseases into Routine Healthcare Practice: A Qualitative Study"

_healthcare, 2023, doi:10.3390/healthcare11172397_

Round 1
Reviewer 1 Report
General comments
=============
The paper “Identifying Factors to Facilitate the Implementation of Decision-Making Tools to Promote Self-Management of Chronic Diseases into Routine Healthcare Practice – A Qualitative Study aims to explore “how self-management-decision tools can be implemented into the routine healthcare settings ensuring an effective use of evidence on SMIs, this study aimed to investigate the implementation factors from the perspective of health care decision-makers and professionals in hospital settings and primary care.” (see p. 2)
Concerning the applied methodology, the manuscript is considered the result of a commendable effort. The authors conducted semi-structured interviews with decision makers (DMs) and health care professionals (HCPs) in Germany and Spain and a focus group discussion with DMs and HCPs from other so-called COMPAR-EU countries between March and October 2022. The interviewees were recruited with the help of an agency, whereas the participants for the focus group discussions were recruited by “partners” (see p. 3). Regarding the methodology, the core contribution is unclear to me due to a lack of a golden thread in the Introduction and detailed information on the COMPAR-EU Project and its aim. Thus, the authors are cordially invited to underline their raison raison d'être in this respect and to structure their Introduction better by giving some background information on the COMPAR-EU Project and its aim. The semi-structured interview guideline's rationale is unclear and lacks a theoretical background. I would invite the authors to expand the rationale, give some background knowledge on what has been known in the research field of interest so far, and explain the COMPAR-EU Project more precisely. What is the aim and the target group of the three mentioned types of decision-making tools (Interactive Summary of Findings tables (iSoF); Evidence to Decision frameworks (EtD); Patient Decision Aids (PtDA), and what are their purposes? Why were the respondents for the interviews recruited in Germany and Spain? Was the research team interested in comparing both countries? They did not focus on cross-country differences because the Results section did not distinguish between both countries. There are also some additional points of criticism which I would raise.
If the authors adequately address the raised concerns, however, the present manuscript could be an interesting contribution to the target journal of Healthcare.
To sum up, concerning the present manuscript, some major weaknesses must be resolved before it can be seriously considered for publication in Healthcare MDPI.
With regard to the writing style, the manuscript, in general, is mostly well-written in terms of style and grammar. In the next step, I would like to elaborate on the weaknesses of the present manuscript in detail and invite the authors to rework it in a thorough revision round.
My comments are described in detail below.
Specific comments
=============
Major and minor comments
---------------------
Please regard the following points as constructive criticism.
Originality/Novelty: Is the question original and well-defined? Do the results provide an advance in current knowledge?
1. The Discussion section mentions that the results build and align with “other bodies of work examining the implementation of decision tools such as patient decision aids or other evidence-based tools (see p. 14). Concerning its novelty/originality, the study aims to explore “how self-management-decision tools can be implemented into the routine healthcare settings ensuring effective use of evidence on SMIs. This study aimed to investigate the implementation factors from the perspective of health care decision-makers and professionals in hospital settings and primary care.” (see p. 2). Including different target groups (DMs and HCPs) is an innovative approach, but unfortunately, the Results section did not differ between both perspectives as the readership might assume before reading. I believe referring to the existing literature in the field in the Discussion section is too late. The authors should include the most important studies in the research field of interest in the Introduction section. In general, the study advances the field, at least in my view. |
Significance: Are the results interpreted appropriately? Are they significant? Are all conclusions justified and supported by the results? Are hypotheses and speculations carefully identified as such?
2. In relation to the manuscript’s significance, the qualitative analysis seems to be conducted thoroughly and appropriately. However, I have to question the raison d'être of the present manuscript under consideration. Due to the lack of clear knowledge of the COMPAR-EU Project, it would have been wise to elaborate on the COMPAR-EU Project in the Introduction. The Special Issue is “Supporting Self-Management in People with Chronic Conditions: Results from the COMPAR-EU Project.” However, considering that MDPI Healthcare is an open-access journal that can be easily found with search terms in search engines, it might not be entirely clear to the readership what the COMPAR-EU Project is. The readership might not be aware that the paper is published, together with others, in this Special Issue. Therefore, too much background information is required on decision-making tools and on the COMPAR-EU Project by the readership to understand the Results section. Thus, I would recommend structuring the Introduction better and identifying a clear research gap from the depiction of what has been known so far on identifying factors to facilitate the implementation of decision-making tools to promote self-management of chronic diseases. The advantages and disadvantages of SMI could also be interesting and described in the Introduction. 3. No hypotheses are included as the authors classify their findings as exploratory. This, however, does not constitute a problem, in my view.
|
Quality of Presentation: Is the article written in an appropriate way? Are the data and analyses presented appropriately? Are the highest standards for the presentation of the results used?
4. I would invite the authors to justify the following in their manuscript: - Why have they chosen to recruit respondents from Germany and Spain? Was it because of cultural differences? - What kind of DMs were invited to participate in the study, and what kind of HDMs were included? - What they meant by “contextualization of the results by the focus group.” 5. Additionally, I would invite the authors to depict a figure with the themes found and arrange them separately for the DMs and the HDMs to visualize differences found in the two groups of interviewees. Differentiation in the Figure along the countries could also make sense. 6. Table 1 reveals the sociodemographic background of the interviewees. I would add two columns in the Table and describe the DMs and HCPS segments separately along the different criteria included in Table 1. 7. I would add another Table 2 describing the focus group discussion participants. 8. I would differentiate the results according to the segments of DMs and HCPs and restructure the Results section in this vein. 9. In total, the references list is expandable. 10. Table 1 is of central importance for the present manuscript. The principle of listing the included studies in the synthesis is not reasonable to me. Is there any order in the list? From my point of view, the studies should be listed according to the authors’ names. Additionally, Table 1 should reveal the country where the study was conducted for all of the studies in a separate column. As I have already stated above, it is not clear to me why the aim of the study was focused on the US and why the systematic review was conducted with studies all over the world. This mismatch is not reasonable. The authors should justify their rationale in this respect. 11. Besides a description of the COMPAR-EU Project on a general level, the three types of decision making tools should be described thoroughly. 12. Maybe a link to the 6-minute-video shown to the respondents could be included in the manuscript? This would maybe enhance the target audience’s comprehension.
|
Scientific Soundness: is the study correctly designed and technically sound? Are the analyses performed with the highest technical standards? Are the data robust enough to draw the conclusions? Are the methods, tools, software, and reagents described with sufficient details to allow another researcher to reproduce the results?
13. Concerning its scientific soundness, the authors should include the semi-structured interview guidelines for the interviews and the focus groups in a multimedia appendix. 14. The whole manuscript is not thoroughly related to the state of the art of literature in the respective research field. It reveals only 24 references. The Introduction should be much more backed up by literature. |
Interest to the Readers: Are the conclusions interesting for the readership of the Journal? Will the paper attract a wide readership, or only interest a few people? (please see the Aims and Scope of the journal).
15. With respect to the readers' interest, the conclusions in general, as well as the practical implications, could be fruitful and interesting. I hope the authors will underline their raison d'être by drawing a clear research gap after having elaborated on what we know on the topic of research of interest. |
Overall Merit: Is there an overall benefit to publishing this work? Does the work provide an advance towards the current knowledge? Do the authors have addressed an important long-standing question with smart experiments?
16. In terms of overall merit, I think that in the event that the authors manage to tackle the problems raised, the manuscript under consideration could make a valuable contribution to the research field. |
English Level: Is the English language appropriate and understandable?
17. I am not a native speaker, but there are some (minor?) grammatical errors in the manuscript as far as I can judge. In general, the manuscrpt is well-written in style, at least in my view. |
The manuscript under consideration can contribute to the research field, but I see some work requiring a diligent revision round. I hope the authors address all raised concerns properly to make this interesting study publishable in Healthcare soon.
Good luck with your research!
I am not a native speaker, but there are some (minor?) grammatical errors in the manuscript as far as I can judge. In general, the manuscrpt is well-written in style, at least in my view.
Reviewer 2 Report
Krah NS et al. performed a qualitative study investigating the factors influencing the implementation and use of self-management interventions decision tools in clinical practice. Indeed, it is a good study but I have some minor points:
· You should include as a limitation that the interviews of the multi-country focus group were conducted in English and not their native language, as in Germans and Spanish.
· Which were your criteria in order to choose participants from Germany and Spain, and not other countries? Please explain.
· Please provide the tool you used including the questions.
Round 2
Reviewer 1 Report
Dear authors!
Thank you for submitting a revised version of the manuscript entitled “Identifying Factors to Facilitate the Implementation of Decision-Making Tools to Promote Self-Management of Chronic Diseases into Routine Healthcare Practice – A Qualitative Study." I appreciate the changes made, and, in my opinion, the quality of the manuscript, which was already high before the former revision round, has improved significantly.
Improvement has been made by elaborating on the COMPAR-EU project and including a textbox in the manuscript to shed light on the entire project. In this respect, you have managed to underline the raison raison d'être of your study. Another significant improvement was made by amending and restructuring some sections and the references list. Thank you for adding the semi-structured interview guideline and justifying the selection of Germany and Spain to choose participants, two issues also raised by Reviewer 2. However, I would not classify the usage of English instead of a native language in a multi-country focus group discussion as a limitation. It is the only feasible way to lead a discussion with different native language participants. Anyway, including more possible limitations of a study that could be raised than fewer might always be a good idea.
Thank you also for adding a new section, 3.5.7, to give an overview of the results by comparing the perspectives of managers vs. health care professionals and underlining similarities and differences, which is very insightful. I also appreciate adding the new Table 2 summarizing the results of the focus group discussions. Thank you also for pointing me and the prospective target readership to the already published protocol paper revealing the overall methodology and research components of the COMPAR-EU project in more detail.
To sum up, you have addressed all of my comments satisfactorily. Very well done! I think the manuscript has improved very much through the revision, and it can make a nice contribution to literature in its present form.
Altogether, from my point of view, there remain only some minor issues that I would like to comment on:
- Please explain abbreviations in full the first time before using them. Thus, in the textbox about the COMPAR-EU project, COPD is mentioned. I would add the abbreviation in brackets the first time it is used in a sentence. The same is true for T2DM or OSF later on. Please also explain NMA in the long-term format before using it as an abbreviation.
- Elaborating on the three types of decision-making tools is helpful for the readers. However, I would recommend explaining what is meant by “based on GRADE” (line 93, p.3).
With regard to my comment no. 10, you are right: This comment is a mistake from my side. I am very sorry.
Overall, however, I think the manuscript will be publishable, and I do not believe that this remainder of criticism constitutes any obstacle to publication. Thus, I will recommend accepting the present manuscript, provided that you still think of considering the remaining issues.
The rest of my comments stated in the first review round have been appropriately addressed, at least from my point of view. I think the manuscript can make a nice contribution to literature in this specific area of research. I hope that I see this interesting manuscript published soon in Healthcare.
Good luck with your research!

Author Response
Dear reviewer,
thank you for the feedback, which helped improve the manuscript!
You raised two more issues:
- "please explain abbreviations in full the first time before using them. Thus, in the textbox about the COMPAR-EU project, COPD is mentioned. I would add the abbreviation in brackets the first time it is used in a sentence. The same is true for T2DM or OSF later on. Please also explain NMA in the long-term format before using it as an abbreviation." --> we have added abbreviations accordingly
- "Elaborating on the three types of decision-making tools is helpful for the readers. However, I would recommend explaining what is meant by “based on GRADE” (line 93, p.3). --> we added an explanation of GRADE.
Best wishes,
Oliver Groene
For more details, please see the revised manuscript.